# Learning Dynamics of Attention:
# Human Prior for Interpretable Machine Reasoning

**Wonjae Kim**
Kakao Corporation
Pangyo, Republic of Korea
dandelin.kim@kakaocorp.com

**Yoonho Lee**
Kakao Corporation
Pangyo, Republic of Korea
eddy.l@kakaocorp.com

## Abstract

Without relevant human priors, neural networks may learn uninterpretable features. We propose **D**ynamics of **A**ttention for **F**ocus **T**ransition (DAFT) as a human prior for machine reasoning. DAFT is a novel method that regularizes attention-based reasoning by modelling it as a continuous dynamical system using neural ordinary differential equations. As a proof of concept, we augment a state-of-the-art visual reasoning model with DAFT. Our experiments reveal that applying DAFT yields similar performance to the original model while using fewer reasoning steps, showing that it implicitly learns to skip unnecessary steps. We also propose a new metric, **T**otal **L**ength of **T**ransition (TLT), which represents the effective reasoning step size by quantifying how much a given model's focus drifts while reasoning about a question. We show that adding DAFT results in lower TLT, demonstrating that our method indeed obeys the human prior towards shorter reasoning paths in addition to producing more interpretable attention maps. Our code is available at https://github.com/kakao/DAFT.

## 1 Introduction

We focus on the task of visual question answering (VQA) [Agrawal et al., 2015], which tests visual reasoning capability by measuring how well a model can answer a question by composing supporting facts from a given image. An example of such a question-image pair from the CLEVR dataset [Johnson et al., 2017a] is shown in Figure 1. One strategy for solving this example is to first find the cube that the question is referring to, and then reporting its color. However, the first step would be unnecessary since all cubes in the image are brown. Questions with such redundancy can be pruned using the complete scene graph. While complete scene graphs are provided in CLEVR, this process is not applicable to real-world images since obtaining their scene graphs is notoriously hard.

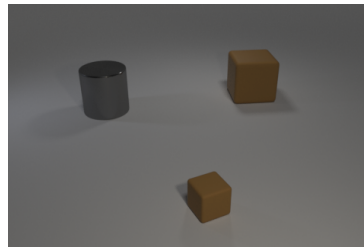

Figure 1: "What color is the cube nearest to the cylinder?" can be answered without knowing the relative location of objects.

The motivation behind training visual reasoning models on the VQA task is to obtain a model that reasons about images similarly to humans. We prefer human-like reasoning because such reasoning is believed to be concise and effective. Conversely, we can say that a model's reasoning is ineffective if it retains and references facts that are irrelevant to the given question, even if its answers are correct. This work is motivated by the question: "How can we measure the degree to which a given model only uses necessary information?"

To this end, we adopt the minimum description length (MDL) principle [Rissanen, 1978], which formalizes Occam's razor and is also a relaxation of Kolmogorov complexity [Kolmogorov, 1963].

This principle states that the best hypothesis for a given data is the one that provides the shortest description of it. The MDL framework offers two benefits: (1) it encourages models to more tightly compress the data, and (2) incentivizes more interpretable models. The first claim comes naturally from the definition of MDL principle since minimum description length is the optimal compression of the data. The inverse relation between interpretability and compression has been demonstrated empirically by numerous works in cognitive neuroscience starting from the work of Hochberg and McAlister [1953] to its modern follow-up studies [Feldman, 2009, 2016].

We thus aim for a VQA method which produces solutions with short description length (in the context of VQA, we also call this a *program*). With the ground-truth program supervision, we can train a model that produces short and effective programs. We aim for an end-to-end learnable reasoning model which produces solutions with short description length. In VQA, such solutions can be seen as learned versions of *explicit* programs, for which we have ground-truth supervision on synthetic datasets such as CLEVR. However such programs are nontrivial to obtain for non-relational questions and can be ill-posed for images with incomplete scene graph. Instead of using the ground-truth program as supervision, we construct a model that *continuously* changes its attention over time, which we experimentally show shifts focus less compared to previous models. This is motivated by experiments [Vendetti and Bunge, 2014] which show that the focus (i.e. attention) of the lateral frontoparietal network on the context changes continuously. Our model, **Dynamics of Attention for Focus Transition** (DAFT), models the infinitesimal change of its attention at each timepoint. Since the resulting attention map is differentiable, it is a continuous funtion over time.

The solution of the initial value problem (IVP) specified by DAFT is a continuous function which specifies the attention map of the model at each point in time. Note that such IVP solutions can be used as a drop-in replacement for any of the discrete attention mechanisms used by previous machine reasoning models. While DAFT is applicable to any attention-based step-wise reasoning model, we applied it to the MAC network [Hudson and Manning, 2018], a state-of-the-art visual reasoning model, to show how this human prior acts in a holistic model. In addition to DAFT, we propose **Total Length of Transition** (TLT), a metric that quantifies the description length of a given attention map, thus measuring the degree to which a model follows the MDL principle. TLT enables a direct quantitative comparison between the quality of reasoning of different models, unlike previous works which only inspected the reasoning of VQA models qualitatively by visualizing attention maps.

This paper is organized as follows. We describe background concepts and their connections to our work in Section 2. We propose DAFT with a detailed explanation of how to adapt DAFT to existing models in Section 3. We present experiments in Section 4, and importantly, we define and measure TLT in Section 4.4. We conclude the paper with future directions in Section 5.

## 2   Background

Our work encompasses multiple disciplines of machine learning including visual question answering, interpretable machine learning, and neural ordinary differential equations. In this section, we summarize each and explain how they are related to our work.

### 2.1   Visual Question Answering

Machine reasoning tasks were proposed to test whether algorithms can demonstrate high-level reasoning capabilities once believed to be only possible for humans [Bottou, 2014]. Given knowledge base $\mathbf{K}$ and task description $\mathbf{Q}$, the model composes supporting facts from $\mathbf{K}$ to accomplish the task described by $\mathbf{Q}$. Visual question answering (VQA) is an instance of a machine reasoning task in the visual domain where $\mathbf{K}$ is an image and $\mathbf{Q}$ is a question about the image ($\mathbf{K}$).

Approaches for solving VQA vary widely on which supervisory signals are given. The usual supervisory signals in VQA comprise images, questions, answers, programs, and object masks. Following Mao et al. [2018], we denote program and object mask supervisions as *additional supervision* and others as *natural supervision*. Natural supervision signals are only signals that all VQA datasets have in common [Agrawal et al., 2015, Krishna et al., 2017, Goyal et al., 2017, Hudson and Manning, 2019], because the additional supervisions are generally hard to acquire.

Given additional supervision, the VQA model can *infer and execute its program on the given scene graph* (i.e. *symbolic* models) [Johnson et al., 2017b]. We refer the reader to Appendix A for further

exposition on models that take this approach. Although symbolic models often employ a neural attention mechanism for program execution (e.g., module networks [Andreas et al., 2016, Hu et al., 2017, Johnson et al., 2017b, Mascharka et al., 2018]), such attention is not necessary if the perfect scene graph can be inferred [Yi et al., 2018].

On the other hand, non-symbolic models, which only use natural supervision, generally all employ some form of attention onto the features of $\mathbf{K}$ from the features of $\mathbf{Q}$ [Xiong et al., 2016, Hudson and Manning, 2018]. Although non-symbolic attention-based models achieve competitive state-of-the-art performance in VQA datasets without additional supervisions (Table 1), no discussions on the effectiveness of its latent program have been made so far. Our work investigates this question by quantitatively measuring the quality of these latent programs and proposes a model that improves on this measure, similarly to how symbolic models are optimized for the effectiveness of their programs.

## 2.2 Human Prior and Interpretability

With the growing demands for interpretable machine learning, attention-based models demonstrated their interpretability by showing their attention map visualizations. However, Ilyas et al. [2019] claimed that without a human prior, neural networks eventually learn *useful but non-robust features* which are highly predictive for the model but not useful for humans. Concurrently, Poursabzi-Sangdeh et al. [2018] and Lage et al. [2018] empirically show how human prior affects the interpretability of the model.

More concretely in VQA, the length of description has no meaning for the model as long as it gets the right answer. For example, [Hudson and Manning, 2018] observed that increasing reasoning step length leaves the model's performance intact (*useful*) but their attention maps became uninterpretable (*non-robust*). To solve this problem, we propose DAFT in Section 3 to embed the human reasoning prior of continuous focus transition in attention-based machine reasoning models.

Another problem is that there exists no method to quantitatively measure the interpretability of attention-based models. This is because interpretability is fundamentally qualitative, and by principle, it can only be measured via a user study. However, user studies cannot scale to large datasets such as CLEVR [Johnson et al., 2017a] GQA [Hudson and Manning, 2019].

Thus we propose TLT as a quantitative and scalable proxy for interpretability, backed with empirical evidence [Hochberg and McAlister, 1953, Feldman, 2009, 2016] in Section 4.4.

## 2.3 Neural Ordinary Differential Equations

Recent work on residual networks [Lu et al., 2017, Haber and Ruthotto, 2017, Ruthotto and Haber, 2018] interpret residual connections as an Euler discretization of a continuous transformation through time. Motivated by this interpretation, Chen et al. [2018] generalized residual networks by using more sophisticated black-box ODE solvers such as `dopri5` [Dormand and Prince, 1980] and proposed a new family of neural networks called neural ordinary differential equations (neural ODEs).

Adaptive-step ODE solvers such as `dopri5` perform multiple function evaluations to adapt their step size, shortening the steps when the gaps between estimations increase and lengthening otherwise. One can find resemblance between adaptive-step ODE solvers and adaptive computation time methods used in recurrent networks [Graves, 2016, Dehghani et al., 2018]. However, as mentioned in [Chen et al., 2018], adaptive-step ODE solvers offer more well-studied, computationally cheap and generalizable rules for adapting the amount of computation. We applied neural ODEs to modeling the infinitesimal change of the model's attention.

Dupont et al. [2019] stated that the homeomorphism of neural ODEs greatly restricts the representation power of the dynamics and show a number of functions which cannot be represented by the family of neural ODEs. They showed that by augmenting the feature space by adding empty dimensions, the dynamics of neural ODEs can be simplified. To show its efficacy, they measured the number of function evaluation (NFE) during training, since complex dynamics requires exponentially many function evaluations while solving IVP. They showed that augmented neural ODEs yield a gradually growing NFE during training while their non-augmented counterpart has an NFE that grows exponentially. We show the connection between our model (DAFT) and augmented neural ODEs in Section 3.

# 3 Dynamics of Attention for Focus Transition

---

**Algorithm 1** Memory Update Procedure of MAC

---

**Input :** current time $t_0$, next time $t_1$, current memory $\mathbf{m}_{t_0}$, contextualized question $\mathbf{cw} \in \mathbb{R}^{L \times d}$, atomic question $\mathbf{q} = [\overleftarrow{\mathbf{cw}_1}, \overrightarrow{\mathbf{cw}_L}]$, knowledge base $\mathbf{K} \in \mathbb{R}^{S \times d}$

**Output :** next memory $\mathbf{m}_{t_1}$

1: $\mathbf{a}_{t_1} = \mathbf{W}^{1 \times d}(\mathbf{W}_{t_1}^{d \times d} \mathbf{q} \odot \mathbf{cw})$      $\triangleright$ get *attention logit* on $\mathbf{cw}$

2: $\mathbf{c}_{t_1} = \sum_{i=0}^{L} \text{softmax}(\mathbf{a}_{t_1})(i) \odot \mathbf{cw}(i)$      $\triangleright$ get *control* vector

3: $\mathbf{rq}_{t_1} = \mathbf{W}^{1 \times d}(\mathbf{W}^{d \times 2d}[\mathbf{W}^{d \times d}\mathbf{K} \odot \mathbf{W}^{d \times d}\mathbf{m}_{t_1}, \mathbf{K}] \odot \mathbf{c}_{t_1})$      $\triangleright$ get *attention logit* on $\mathbf{K}$

4: $\mathbf{r}_{t_1} = \sum_{i=0}^{S} \text{softmax}(\mathbf{rq}_{t_1})(i) \odot \mathbf{K}(i)$      $\triangleright$ get *information* vector

5: $\mathbf{m}_{t_1} = \mathbf{W}^{d \times 2d}[\mathbf{r}_{t_1}, \mathbf{m}_{t_0}]$      $\triangleright$ get *memory* vector

---

**The MAC Network**   We briefly review the MAC network [Hudson and Manning, 2018]. It consists of three subunits (control, read, and write) which rely on each other to perform visual reasoning. Algorithm 1 describes how the MAC network updates its memory vector given its inputs. Given initial memory vector $\mathbf{m}_0$, it performs a fixed number ($T$) of iterative memory updates to produce the final memory vector $\mathbf{m}_T$. MAC infers answer logits by processing the concatenation of $\mathbf{q}$ and $\mathbf{m}_T$ through a 2-layer classifier : $\mathbf{W}^{1 \times d}(\mathbf{W}^{d \times 2d}[\mathbf{q}, \mathbf{m}_T])$[1]. The original work optionally considers additional structures inside the write unit. Unlike the description in the original paper, previous control $\mathbf{c}_{t-1}$ is not used when computing the current control $\mathbf{c}_t$ in the official implementation[2]. Please refer the original paper [Hudson and Manning, 2018] for the details.

---

**Algorithm 2** Memory Update Procedure of DAFT MAC

---

**Input :** current time $t_0$, next time $t_1$, current memory $\mathbf{m}_{t_0}$, contextualized question $\mathbf{cw} \in \mathbb{R}^{L \times d}$, atomic question $\mathbf{q} = [\overleftarrow{\mathbf{cw}_1}, \overrightarrow{\mathbf{cw}_L}]$, knowledge base $\mathbf{K} \in \mathbb{R}^{S \times d}$, current attention logit $\mathbf{a}_{t_0}$

**Output :** next memory $\mathbf{m}_{t_1}$, next attention logit $\mathbf{a}_{t_1}$

1: **def** f$(\mathbf{a}_t, t)$:      $\triangleright$ **Define DAFT**

2:      **return** $\mathbf{W}^{1 \times (d+1)}[\mathbf{W}^{d \times (d+1)}[t, \mathbf{q}] \odot \mathbf{cw}, \mathbf{a}_t]$      $\triangleright$ compute $\frac{d\mathbf{a}_t}{dt}$

3: $\mathbf{a}_{t_1} = \mathbf{a}_{t_0} + \int_{t_0}^{t_1} f(\mathbf{a}_t, t)dt = \text{ODESolve}(\mathbf{a}_t, f, t_0, t_1)$      $\triangleright$ Solve IVP using DAFT

4: $\mathbf{c}_{t_1} = \sum_{i=0}^{L} \text{softmax}(\mathbf{a}_{t_1})(i) \odot \mathbf{cw}(i)$

5: $\mathbf{rq}_{t_1} = \mathbf{W}^{1 \times d}(\mathbf{W}^{d \times 2d}[\mathbf{W}^{d \times d}\mathbf{K} \odot \mathbf{W}^{d \times d}\mathbf{m}_{t_0}, \mathbf{K}] \odot \mathbf{c}_{t_1})$

6: $\mathbf{r}_{t_1} = \sum_{i=0}^{S} \text{softmax}(\mathbf{rq}_{t_1})(i) \odot \mathbf{K}(i)$

7: $\mathbf{m}_{t_1} = \mathbf{W}^{d \times 2d}[\mathbf{r}_{t_1}, \mathbf{m}_{t_0}]$

---

**The DAFT MAC Network**   We now introduce Dynamics of Attention for Focus Transition (DAFT) and its application to MAC; we call this augmented MAC model as DAFT MAC.

Algorithm 2 shows the memory update procedure of DAFT MAC and the definition of DAFT in full detail. We colored the differences in Algorithm 1 and Algorithm 2. We point out that DAFT can just as easily be applied to any other memory-augmented model by replacing discrete attention with a neural ODE as we have done in Algorithm 2.

Unlike MAC, the memory update procedure of DAFT MAC requires the previous attention logit, meaning we need to define the initial attention logit. We use a zero vector as the initial attention logit $\mathbf{a}_0$ to produce uniformly distributed attention weight, assuming the model's focus distributed evenly at the start of reasoning.

Figure 2 shows the difference between MAC and DAFT MAC graphically. While MAC has no explicit connection between adjacent logits, DAFT MAC computes the next attention logit by solving the IVP starting from the current attention logit. Note that the actual attention weight is the softmax-ed value of attention logits. Since softmax computes the size of a logit relative to other logits, small

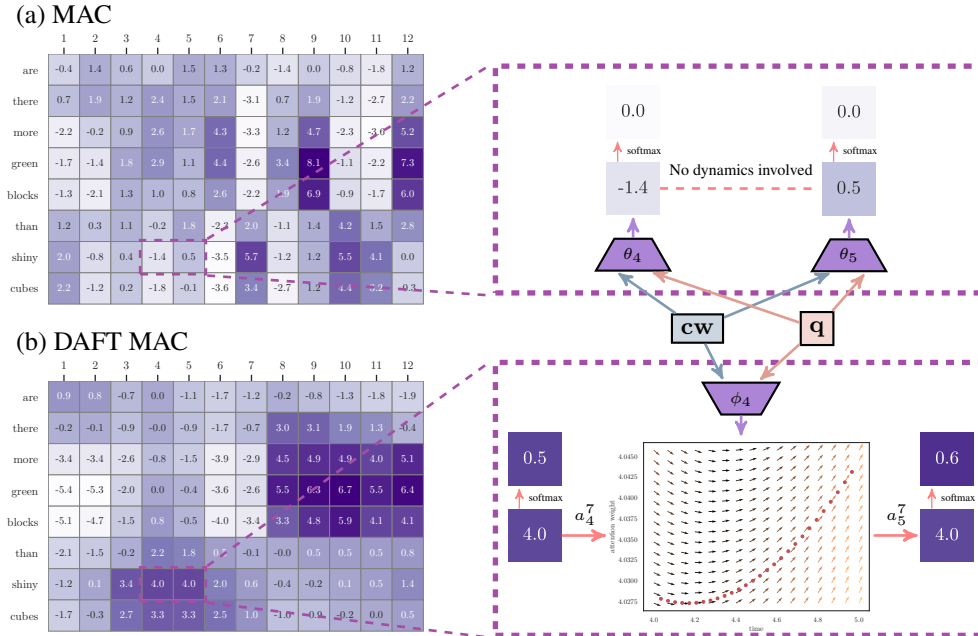

Figure 2: A graphical description of how attention logits change in MAC and DAFT MAC for an example in the CLEVR dataset. The question is *"are there more green blocks than shiny cubes?"*. Attention logits maps of 12-step (a) MAC and (b) DAFT MAC are shown. The right side shows a magnified view of a single step of attention shift on the word *shiny*.

changes in attention logit can result in a large difference in the attention weight (See Figure 4 for a visualization of the attention weight).

**Connection to Augmented Neural ODEs**   As shown in Figure 2, every token **cw** and its question **q** acts as a condition on the dynamics. Empirically, we found that the conditionally generated ODE dynamics do not suffer from number of function evaluations (NFE) explosion while solving IVP until the end of training (see Figure 10 in the appendix for more details on NFE). This is remarkable since the VQA is incomparably more complex than the toy problems treated in previous works. We thus argue that these conditional ODE dynamics are another form of augmentation for neural ODEs as it differs from the previous unconditioned neural ODEs [Chen et al., 2018, Dupont et al., 2019].

**Alternative Ways to Restrict Focus Transition**   Besides DAFT, we tested two simple alternatives to restrict the model's transition of attention. The first is to introduce a residual connection at each attention step, which is equivalent to DAFT using a single-step Euler solver during training. We observed significant drops in accuracy, and attention maps of this model deffered all transitions to the last few steps. We attribute this phenomenon to this residual model having insufficient expressive power compared to the complex visual information being incorporated at each step. Our second baseline is to add the TLT itself to objective function with Lagrange multiplier $\lambda$. This model significantly harmed performance for every $\lambda$ in the wide range we tested.

## 4   Experiments

We conducted our experiments on the CLEVR[3] [Johnson et al., 2017a] and GQA[4] [Hudson and Manning, 2019] datasets. For brevity we put the results from GQA dataset in the Appendix C.

To evaluate the efficacy of DAFT, we conducted experiments on two different criteria: performance (accuracy and run-time) and interpretability. For a fair comparison, we used the same hyperparameters

as the original MAC network [Hudson and Manning, 2018] and closely followed their experimental setup. The only difference from the original MAC network is in the computation of attention logits and control vectors (highlighted in purple in Algorithm 2). We list implementation details in Appendix B.

## 4.1 CLEVR Dataset

Table 1: Accuracies on the CLEVR dataset of baselines with various additional annotation types (**P** for program and **M** for object mask annotation) and our model. $D$ denotes depth of the inferred program. $\triangle$ means that additional annotation is implicitly provided through the pretrained object detector such as Mask R-CNN.

| Model | Anno. P | M | # Step | Avg. | Count | Exist | Cmp. Num. | Query Attr. | Cmp. Attr. |
|---|---|---|---|---|---|---|---|---|---|
| Human [Johnson et al., 2017a] | – | – | – | 92.6 | 86.7 | 96.6 | 86.5 | 95.0 | 96.0 |
| NMN [Andreas et al., 2016] | O | X | $D$ | 72.1 | 52.5 | 79.3 | 72.7 | 79.0 | 78.0 |
| N2NMN [Hu et al., 2017] | O | X | $D$ | 88.8 | 68.5 | 85.7 | 84.9 | 90.0 | 88.8 |
| IEP [Johnson et al., 2017b] | O | X | $D$ | 96.9 | 92.7 | 97.1 | 98.7 | 98.1 | 98.9 |
| DDRprog [Suarez et al., 2018] | O | X | $D$ | 98.3 | 96.5 | 98.8 | 98.4 | 99.1 | 99.0 |
| TbD [Mascharka et al., 2018] | O | X | $D$ | 99.1 | 97.6 | 99.2 | 99.4 | 99.5 | 99.6 |
| NS-VQA [Yi et al., 2018] | O | O | $D$ | 99.8 | 99.7 | 99.9 | 99.9 | 99.8 | 99.8 |
| NS-CL [Mao et al., 2018] | X | $\triangle$ | $D$ | 98.9 | 98.2 | 99.0 | 98.8 | 99.3 | 99.1 |
| RN [Santoro et al., 2017] | X | X | 1 | 95.5 | 90.1 | 97.8 | 93.6 | 97.1 | 97.9 |
| FiLM [Perez et al., 2018] | X | X | 4 | 97.6 | 94.5 | 99.2 | 93.8 | 99.2 | 99.0 |
| MAC [Hudson and Manning, 2018] | X | X | 12 | 98.9 | 97.2 | 99.5 | 99.4 | 99.3 | 99.5 |
| DAFT MAC (Ours) | X | X | **4** | 98.9 | 97.2 | 99.5 | 98.3 | 99.6 | 99.3 |

CLEVR dataset was proposed to evaluate the visual reasoning capabilities of a model. CLEVR includes five supervisory signals: images, questions, answers, programs, and object masks (in addition to ground-truth scene graphs). Images in CLEVR are synthetic scenes containing objects with various attributes: size, material, color, shape. Each image has multiple questions with corresponding answers to test relational and non-relational visual reasoning abilities.

We provide a survey of previous models for CLEVR in Table 1, showing the accuracy by question type in addition to what additional supervision is given to the model. In total, CLEVR has 700K questions for training and 150K questions for validation and test split. All accuracies and TLT measured in the following sections were evaluated on the 150K validation set.

## 4.2 Performance

We re-implemented MAC along with DAFT MAC. We consider a wide range of numbers of steps between 2 and 30, and trained each pair of (method, step number) five times using different random seeds for thorough verification. As shown in Figure 3, the accuracy of DAFT MAC outperforms that of the original MAC for fewer reasoning steps ($2 \sim 6$), and the two methods are roughly tied for larger reasoning steps. Hudson and Manning [2018] reported that MAC achieves its best accuracy (98.9%) at step size 12; DAFT MAC reaches equal performance with step size 4. In our experiments, MAC and DAFT MAC both reach 99.0% accuracy at step size 8. Increasing step size beyond 8 results in practically the same performance while requiring more computation; in our experiments, 12-step took $\sim$28% more time compared to 8-step.

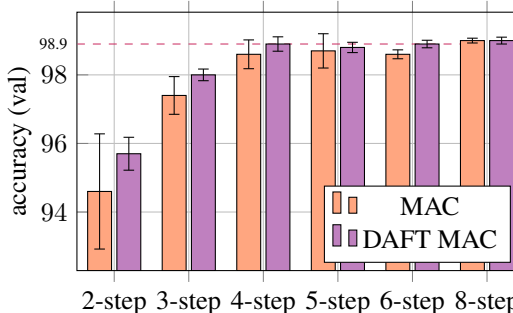

Figure 3: Comparison of CLEVR mean accuracy and 95% confidence interval ($N = 5$) between MAC and DAFT MAC with varying reasoning steps.

The fact that the accuracy of DAFT MAC does not increase when increasing the reasoning step beyond four suggests that four reasoning steps are sufficient for the CLEVR dataset. We provide more justification for this claim in Section 4.4 by quantifying the effective number of reasoning steps in each model.

Table 2: Run-time analysis of MAC and DAFT MAC with various ODE solvers.

| Model | MAC | DAFT MAC | DAFT MAC | DAFT MAC |
| Solver | - | Euler | Runge-Kutta 4th order | Dormand-Prince |
| --- | --- | --- | --- | --- |
| Accuracy | $98.6 \pm 0.2$ | $98.7 \pm 0.2$ | $98.9 \pm 0.2$ | $98.9 \pm 0.2$ |
| TLT | $2.06 \pm 0.15$ | $1.76 \pm 0.07$ | $1.62 \pm 0.06$ | $1.62 \pm 0.06$ |
| Time (ms) | $153.7 \pm 3.8$ (1x) | $167.9 \pm 1.7$ (1.09x) | $189.7 \pm 1.9$ (1.23x) | $365.5 \pm 12.5$ (2.37x) |

We additionally ran a more detailed run-time analysis. We measured the accuracy, TLT, and time for inferring a batch of 64 question-image pairs, using various ODE solvers *during evaluation* of five different 4-step DAFT MAC. We used two fixed-step solvers (Euler method and Runge-Kutta 4th order method with 3/8 rule) and one adaptive-step solver (Dormand-Prince method) that we used during training. We found that during evaluation, Runge-Kutta solves all the dynamics generated from CLEVR dataset. Note that even the simplest Euler method results in higher accuracy and lower TLT compared to vanila MAC.

## 4.3 Interpretability

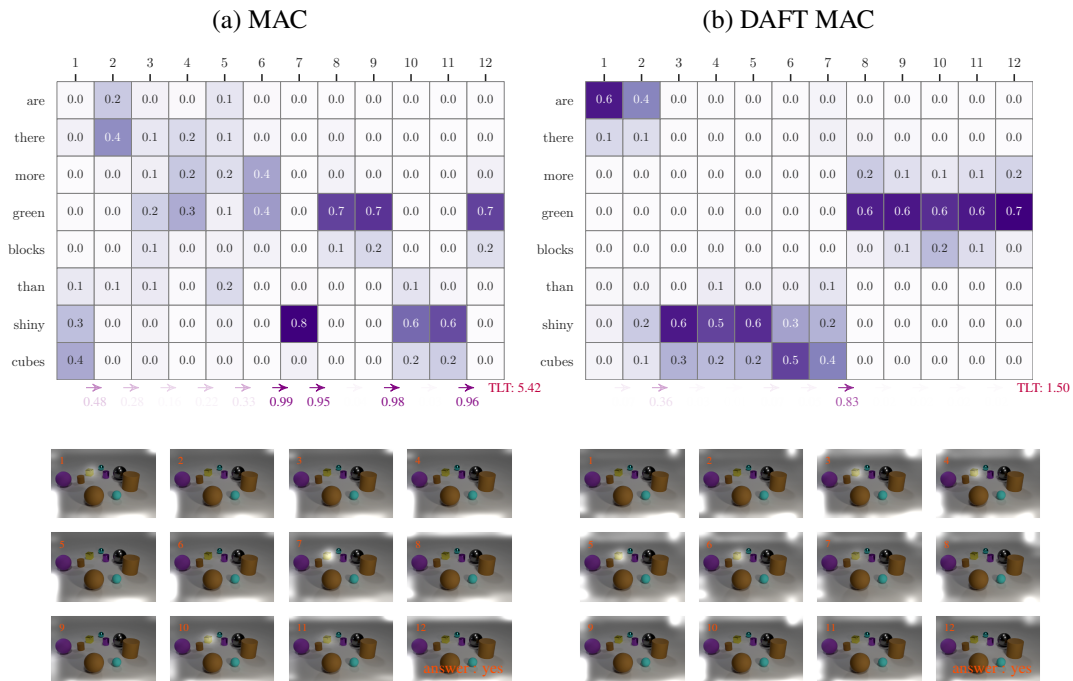

Figure 4: Attention maps for the question *"Are there more green blocks than shiny cubes?"* and its accompanying image, the same data used to show attention logit map in Figure 2. (a) and (b) shows the actual softmax-ed textual and visual attention map which used to acquire the control vector and the information vector in MAC and DAFT MAC, respectively.

Many attention-based machine reasoning models put emphasis on the interpretability of the attention map [Lu et al., 2016, Kim et al., 2018, Hudson and Manning, 2018]. Indeed, the attention map is a great source of interpretation since it points to specific temporal and spatial points helping our mind to interpret the observation. In Figure 4, we compared the qualitative visualization of attention maps for MAC and DAFT MAC. One can see that DAFT's human prior is beneficial for interpretation in several ways:

**Chunking**   Compared to MAC, DAFT MAC produces more clustered and chunky attention maps. The question *"Are there more green blocks than shiny cubes?"* contains two noun phrases (NP), *more green blocks* and *shiny cubes*, when parsed to (S Are there (NP (ADJP (ADVP more) green) blocks) (PP than (NP shiny cubes))). In this simple case, an ideal solver would only see each NP once to solve the problem. In Figure 4, MAC distributes its attention to multiple temporally distant position to retrieve information while DAFT MAC distributes its attention to the chunks which are the same number as the question's NPs.

**Consistency**   The attention maps produced by DAFT MAC presents a consistent progression of focus. We observed that DAFT MACs initialized with different seeds shares the order of transition. While the learned attention map of MAC varies greatly across different initializations, DAFT MAC consistently attends to *shiny cube* first and then afterwards to *more green blocks* (see Figure 12 and Figure 13 in the appendix for the clear distinction).

**Interpolation**   Since the solution of IVP can yield an attention map for any given point in time, we can easily interpolate the attention maps in-between two adjacent steps. See Figure 14 in the appendix for a visualization of these interpolated maps. Note that although we visualized the interpolation with the sampling rate of 20 due to limited space, this rate can go infinitely high since DAFT is continuous in time. This interpolation differs from simple linear interpolation since DAFT has non-linear dynamics.

### 4.4   Total Length of Transition

To mesure the description length of a given attention map, we first define the *length* of the map. Recall that the attention map is a categorical distribution over input tokens. A simple example of quantifying the distance of such a map is to choose the word to which the model focused on most at each time step, and measure the number of times this shifted. For example, the attention map of DAFT MAC in Figure 4 can be simplified as ["are", "shiny", "cubes", "green"] and the map of MAC as ["cubes", "there", "green, "than", "green", "shiny", "green", "shiny", "green"]. If we measure the length in this way, the lengths become $4$ and $9$, respectively.

However, since we have more finer information than just gathering tokens with maximum values, we can employ probabilistic measures of distances. The distances will generally follow the simple discrete measurement and can measure more precise length of given attention map. Thus we use the Jensen-Shannon divergence [Lin, 1991] to measure the amount of shift between attention maps throughout reasoning. We chose the Jensen-Shannon divergence because it is bounded ($JSD(P||Q) \in [0, 1]$).

**Definition 1**  *Length of Transition (LT)*
*Let $\mathbf{p}_t \in \mathbb{R}^S$ be the attention probability for time $t = 1, \ldots, T$. The Length of Transition (LT) at time $t$ is defined as:*

$$LT(t) = JSD(\mathbf{p}_t||\mathbf{p}_{t+1}) = \frac{1}{2} \sum_{s=1}^{S} p_t^s \cdot \log_2 \frac{2 \cdot p_t^s}{p_t^s + p_{t+1}^s} + p_{t+1}^s \cdot \log_2 \frac{2 \cdot p_{t+1}^s}{p_t^s + p_{t+1}^s} \tag{1}$$

*where $p_t^s$ is the s-th element of $\mathbf{p}_t$.*

We further define total length of transition (TLT) as TLT $= \sum_{i=1}^{T} LT(i)$ [5]. In default, TLT is bounded by $T - 1$, and if TLT considers $LT(0)$, it is bounded by $T$. One can concatenate uniformly distributed attention to $\mathbf{a}$ as a starting attention $\mathbf{a}_0$ to get $LT(0)$. We do not use $LT(0)$ when calculating TLT throughout this paper, making it bounded by $T - 1$. Furthermore, we argue that a model with low TLT is more likely to produce consistent attention maps across different initializations since TLT imposes an upper bound on the amount the model's attention can change. We denoted LTs and TLT for MAC and DAFT MAC at the below of attention maps in Figure 4.

Figure 5 shows the TLT values of MAC and DAFT MAC. When the number of reasoning steps increases, the TLT of DAFT MAC is relatively unchanged while that of MAC increases with step number. This result supports the qualitative result shown before and demonstrates that DAFT MAC

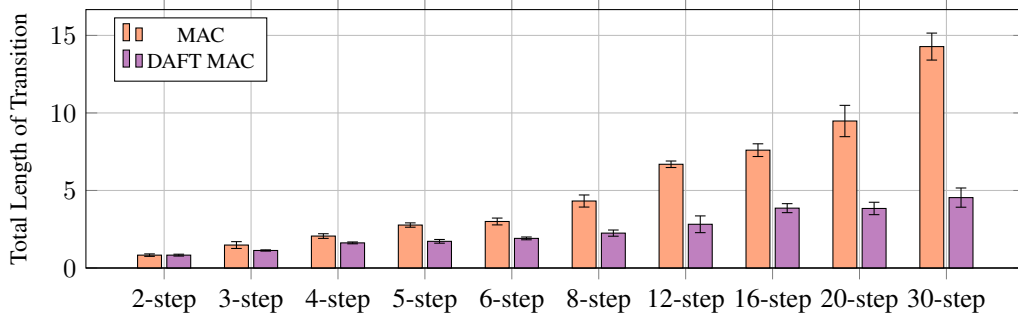

Figure 5: Comparison of CLEVR mean TLT and its 95% confidence interval ($N = 5$) between MAC and DAFT MAC with varying reasoning steps.

consistently results in simplified reasoning paths across the whole dataset, rather than only in a few cherry-picked examples. In Section 4.2, we have argued that the 4-step is enough for solving CLEVR. In Figure 5, one can see that step-wise growth reaches its maximum in 4-step (for clear view, see Figure 11 in the appendix), implying that the model requires more space to navigate its focus when the step size is smaller than four.

Figure 6 shows how much TLT each question type yields. Since TLT grows with the size of the reasoning step, we employed a relative value of TLT to normalize this value across different numbers of training steps. Relative TLT is defined as $TLT_t(question\_type)/TLT_t$, where $t$ ranges over steps in Figure 5. The fact that each question type's relative TLT has the same order within both MAC and DAFT MAC substantiates TLT's ability to measure reasoning complexity regardless of the specific architecture.

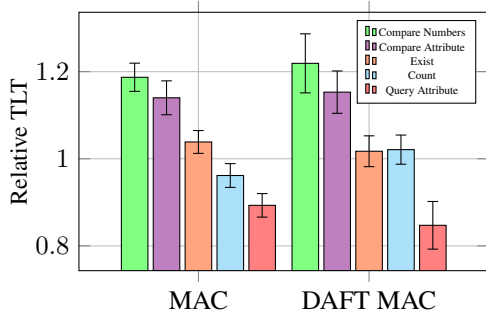

Figure 6: Comparison of relative TLT mean accuracy and its 95% confidence interval ($N = 50$) with varying question type.

Question types *Compare Numbers* and *Compare Attribute* had higher TLT than other question types. This is expected since such comparative questions involve more NP chunks than other question types. When we shrank the step size from four to two, the accuracy of *Query Attribute* question type was pretty much unharmed ($99.6 \rightarrow 99.3$ in DAFT MAC and $99.6 \rightarrow 97.5$ in MAC) while that of other question types significantly dropped. This is supported by the fact that *Query Attribute* question type had lowest TLT, meaning the question type is solvable using a small number of steps.

## 5  Conclusion

We have proposed Dynamics of Attention for Focus Transition (DAFT), which embeds the human prior of continuous focus transition. In contrast to previous approaches, DAFT learns the dynamics in-between reasoning steps, yielding more interpretable attention maps. When applied to MAC, the state-of-the-art among models that only use natural supervision, DAFT achieves the same performance while using $1/3$ the number of reasoning steps. In addition, we proposed a novel metric called Total Length Transition (TLT). Following the minimum description length principle, TLT measures how good the model is on planning effective, short reasoning path (latent program), which is directly related to the interpretability of the model.

Next on our agenda includes (1) extending DAFT to other tasks where performance and interpretability are both important to develop a method to balance between the two criteria, and (2) investigating what other values TLT can serve as a proxy for.

## Footnotes

[1]We omit biases and nonlinearities for brevity.

[2]https://github.com/stanfordnlp/mac-network/blob/master/configs/args.txt

[3] https://cs.stanford.edu/people/jcjohns/clevr/

[4] https://cs.stanford.edu/people/dorarad/gqa/about.html

[5]This is quite similar to the length of the prequential (online) code of Blier and Ollivier [2018], with the difference that theirs is a sum of negative log probabilities instead of a JS divegence

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
