[Supplementary Material]

# Appendix for Learning Dynamics of Attention: Human Prior for Interpretable Machine Reasoning

**Wonjae Kim**
Kakao Corporation
Pangyo, Republic of Korea
dandelin.kim@kakaocorp.com

**Yoonho Lee**
Kakao Corporation
Pangyo, Republic of Korea
eddy.l@kakaocorp.com

## A    Reasoning with Additional Supervision

In this section, we detail the additional supervisions and taxonomize previous studies on visual reasoning according to each additional supervision signal used.

**Program**    A (functional) program is a set of logical functions that can be executed on an image's scene graph. Programs are a valuable supervision for VQA models since it enables the model to convert a natural language question into excutable functions [Andreas et al., 2016, Hu et al., 2017, Mascharka et al., 2018, Yi et al., 2018]. Such composition of functions provides far better interpretability than that of models which interact with raw sensory data (i.e., natural supervisions).

However, since programs are generally disentangled from scene graphs, their supervision can keep the model from learning to skip unnecessary steps as we discussed in the introduction. This problem can be handled by only giving the model *optimal* programs. Synthetically generated dataset such as CLEVR [Johnson et al., 2017] pruned out suboptimal programs like shown in Figure 1 by inspecting their related scene graph. However, even such pruning requires a perfect scene graph, which is not available in real-world datasets. Therefore the programs provided in real-world VQA datasets such as GQA [Hudson and Manning, 2019] are inherently suboptimal since they are generated from approximate scene graphs.

**Object Mask**    The other thing that makes solving VQA hard is that each node in the scene graph corresponds to a different group of pixels. Object masks help the model locate nodes (i.e., objects) of the scene graph before the reasoning steps (i.e., program execution). Well annotated masks relieve the model from finding objects and allow them to solely concentrate on reasoning.

Yi et al. [2018] generated object masks for the CLEVR dataset using its scene graph, and used them as supervisory signals for inferring the scene graph. In the specific setup of CLEVR, they achieved near perfect performance (99.8%). Such result shows that object masks and their corresponding inference module such as Mask R-CNN [He et al., 2017] are enough to build an exact scene graph in synthetically generated images. However, annotating object masks and their scene graphs for real-world images is still an open problem.

## B    Implementation Details

To solve the initial value problem of the ODE, we used torchdiffeq [Chen et al., 2018]. For designing and accelerating the computation graph of the model, we used pytorch 1.0.1 [Paszke et al., 2017] with CUDA 9.2 on an Nvidia V100 GPU. Every experiment was performed with five different initial seeds by fixing the inital seed with manual_seed() for python, pytorch, and numpy.

We used the Adam optimizer [Kingma and Ba, 2014] with learning rate 1e-4 for all experiments, and halved the learning rate whenever the validation accuracy stopped improving for more than one epoch.

We trained using batches of 64 training data and terminated training when the learning rate went under 1e-7. The size of all hidden dimensions was fixed to 512 except for the word embedding layer, which was 300. All weights for the affine transformation were initialized with xavier initialization [Glorot and Bengio, 2010], and word embeddings were initialized to random vectors using a uniform distribution following the settings of MAC.

# C   Experiments on GQA

(a) MAC

| | 1 | 2 | 3 | 4 | 5 | 6 | 7 | 8 | 9 | 10 | 11 | 12 |
|---|---|---|---|---|---|---|---|---|---|---|---|---|
| do | 0.1 | 0.0 | 0.2 | 0.2 | 0.1 | 0.0 | 0.0 | 0.7 | 0.0 | 0.0 | 0.5 | 0.0 |
| you | 0.1 | 0.0 | 0.1 | 0.2 | 0.1 | 0.0 | 0.0 | 0.2 | 0.0 | 0.0 | 0.3 | 0.0 |
| see | 0.2 | 0.0 | 0.2 | 0.4 | 0.1 | 0.0 | 0.0 | 0.0 | 0.0 | 0.0 | 0.0 | 0.0 |
| either | 0.4 | 0.0 | 0.1 | 0.1 | 0.1 | 0.0 | 0.0 | 0.0 | 0.0 | 0.0 | 0.0 | 0.0 |
| any | 0.2 | 0.0 | 0.0 | 0.1 | 0.1 | 0.0 | 0.0 | 0.0 | 0.0 | 0.0 | 0.0 | 0.0 |
| mirrors | 0.0 | 0.0 | 0.0 | 0.0 | 0.3 | 0.7 | 0.0 | 0.0 | 0.7 | 0.5 | 0.0 | 0.0 |
| or | 0.0 | 0.0 | 0.3 | 0.0 | 0.0 | 0.2 | 0.0 | 0.0 | 0.1 | 0.5 | 0.0 | 0.0 |
| benches | 0.0 | 1.0 | 0.1 | 0.0 | 0.2 | 0.0 | 1.0 | 0.0 | 0.2 | 0.0 | 0.0 | 1.0 |

TLT: 7.19
1.00  0.68  0.29  0.30  0.32  0.79  0.99  0.97  1.23  0.78  0.85

(b) DAFT MAC

| | 1 | 2 | 3 | 4 | 5 | 6 | 7 | 8 | 9 | 10 | 11 | 12 |
|---|---|---|---|---|---|---|---|---|---|---|---|---|
| do | 0.0 | 0.0 | 0.2 | 0.1 | 0.0 | 0.0 | 0.0 | 0.0 | 0.0 | 0.0 | 0.0 | 0.0 |
| you | 0.1 | 0.1 | 0.2 | 0.1 | 0.0 | 0.0 | 0.0 | 0.0 | 0.0 | 0.0 | 0.0 | 0.0 |
| see | 0.3 | 0.5 | 0.4 | 0.4 | 0.0 | 0.0 | 0.0 | 0.0 | 0.0 | 0.0 | 0.0 | 0.0 |
| either | 0.2 | 0.2 | 0.1 | 0.1 | 0.0 | 0.0 | 0.0 | 0.0 | 0.0 | 0.0 | 0.0 | 0.0 |
| any | 0.2 | 0.1 | 0.0 | 0.2 | 0.1 | 0.0 | 0.0 | 0.0 | 0.0 | 0.0 | 0.0 | 0.0 |
| mirrors | 0.0 | 0.0 | 0.0 | 0.0 | 0.7 | 0.7 | 0.0 | 0.2 | 0.0 | 0.1 | 0.0 | 0.2 |
| or | 0.0 | 0.0 | 0.0 | 0.0 | 0.0 | 0.0 | 0.0 | 0.0 | 0.0 | 0.0 | 0.0 | 0.0 |
| benches | 0.0 | 0.0 | 0.0 | 0.0 | 0.1 | 0.3 | 1.0 | 0.8 | 1.0 | 0.9 | 1.0 | 0.8 |

TLT: 1.68
0.67  0.45

Figure 7: A graphical description of how attention maps change in MAC and DAFT MAC for a GQA example. The given question is *"do you see either any mirrors or benches?"*. Attention maps of 12-step (a) MAC and (b) DAFT MAC are shown. In both textual and visual, DAFT MAC's attention changes from mirrors to benches smoothly.

Figure 8: Comparison of GQA mean TLT and its 95% confidence interval ($N = 5$) between MAC and DAFT MAC with varying reasoning steps.

The GQA [Hudson and Manning, 2019] dataset is a real-world VQA dataset where all questions are generated compositionally by injecting scene graph information into question templates. Although it shares most details with CLEVR dataset, its questions are far less complex than that of CLEVR. Creating complex questions in GQA is challenging because while each of its images typically contain objects from many different classes, the number of objects for each class is small.

On the other hand, an image in the CLEVR dataset contains few classes of objects, but the number of objects per class is large; this enables CLEVR to make compositionally complex questions. In Figure 7, we compare the attention maps of MAC and DAFT MAC just as we did in Figure 4.

Figure 8 shows the TLTs of the two methods on the GQA dataset while varying reasoning steps. Compared with Figure 5, one can see that the TLT of GQA is far less than that of CLEVR. This confirms that questions of GQA is indeed less complex than that of CLEVR.

Figure 9: Comparison of overall GQA mean accuracy and its 95% confidence interval ($N = 5$) between MAC and DAFT MAC with varying reasoning steps.

Figure 9 shows accuracies of MAC and DAFT MAC when evaulated on the GQA dataset. As shown in the figure, there is not much difference in accuracy across step sizes for both MAC and DAFT MAC. It tells us that 2-step is roughly enough, as 4-step MAC in CLEVR did. We would like to note that we observed many runs that achieve over 54% validation accuracy (which matches with the accuracy Hudson and Manning [2019] reported) for some periods. However, the accuracies fall shortly after the peak and converged in reported accuracies. Since we reported all accuracies and TLTs with the model which passed through full training session as mentioned in Appendix B throughout the paper, we report the results of GQA in the same manner.

## D Additional Figures

Figure 10: Growth of the Number of Function Evaluation (NFE) for 4-step DAFT MAC as training progresses. Mean value and 95% confidence interval ($N = 5$) are denoted as line and gradation.

Figure 11: Mean growth of TLT that starts from 2-step. Bars denote arithmetic mean value of given interval. For example, the bar at 12 represents $\frac{TLT_{12}-TLT_2}{10}$. The figure is linked with Figure 5.

**Top-left attention map**

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

Figure 13: Attention maps from the other four 12-step DAFT MACs initialized with different seeds, distributed over question *"Are there more green blocks than shiny cubes?"*. All of them perform similiarly to the model used in Figure 4 in terms of CLEVR validation accuracy.

|  | 1 | 2 | 3 | 4 | 5 | 6 | 7 | 8 |
|---|---|---|---|---|---|---|---|---|
| how | 0.0 | 0.0 | 0.0 | 0.0 | 0.0 | 0.0 | 0.0 | 0.0 |
| many | 0.0 | 0.0 | 0.0 | 0.0 | 0.4 | 0.2 | 0.0 | 0.0 |
| objects | 0.0 | 0.0 | 0.0 | 0.0 | 0.1 | 0.3 | 0.5 | 0.2 |
| are | 0.0 | 0.0 | 0.0 | 0.0 | 0.0 | 0.0 | 0.0 | 0.0 |
| balls | 0.0 | 0.0 | 0.0 | 0.0 | 0.0 | 0.3 | 0.4 | 0.0 |
| behind | 0.0 | 0.0 | 0.0 | 0.0 | 0.0 | 0.0 | 0.0 | 0.0 |
| the | 0.5 | 0.1 | 0.1 | 0.0 | 0.0 | 0.0 | 0.0 | 0.0 |
| big | 0.1 | 0.3 | 0.2 | 0.0 | 0.0 | 0.0 | 0.0 | 0.0 |
| brown | 0.2 | 0.5 | 0.5 | 0.0 | 0.0 | 0.0 | 0.0 | 0.0 |
| object | 0.0 | 0.0 | 0.0 | 0.0 | 0.0 | 0.0 | 0.0 | 0.0 |
| or | 0.0 | 0.0 | 0.0 | 0.0 | 0.0 | 0.0 | 0.0 | 0.0 |
| blue | 0.0 | 0.0 | 0.0 | 0.0 | 0.1 | 0.0 | 0.0 | 0.3 |
| matte | 0.0 | 0.0 | 0.0 | 0.0 | 0.0 | 0.0 | 0.0 | 0.1 |
| balls | 0.0 | 0.0 | 0.0 | 0.0 | 0.0 | 0.0 | 0.0 | 0.1 |
| behind | 0.0 | 0.0 | 0.0 | 0.0 | 0.0 | 0.0 | 0.0 | 0.0 |
| the | 0.0 | 0.0 | 0.0 | 0.2 | 0.0 | 0.0 | 0.0 | 0.0 |
| cyan | 0.0 | 0.0 | 0.0 | 0.5 | 0.0 | 0.0 | 0.0 | 0.0 |
| matte | 0.0 | 0.0 | 0.0 | 0.1 | 0.0 | 0.0 | 0.0 | 0.0 |
| ball | 0.0 | 0.0 | 0.0 | 0.0 | 0.0 | 0.0 | 0.0 | 0.0 |

Figure 14: Interpolation in-between steps. Since the solution of IVP is a continuous function of time, we can get a attention map for any given intermediate time value. This fact enables infinitely fine-grained interpolation. Also note that this is not a linear interpolation, see how the attention on *many* reaches a maximum around 5.2 instead of on either end.

Figure 15: Attention maps of 8-step MAC, distributed over question *"How many objects are balls behind the big brown object or blue matte balls behind the cyan matte ball?"*. This model achieves 99% CLEVR validation accuracy.

Figure 16: Attention maps of 8-step DAFT MAC, distributed over question *"How many objects are balls behind the big brown object or blue matte balls behind the cyan matte ball?"*. This model achieves 99% CLEVR validation accuracy.

Figure 17: Attention maps of DAFT MAC with 2 to 6 steps, distributed over the very long question *"How many objects are either big things that are on the left side of the small brown metallic cube or rubber things that are on the right side of the tiny blue rubber thing?"*. Note that these five models are seperately initialized and thus have totally *different* parameters. The order of transition is unchanged among these completely separate models with different expressive power.

Figure 18: Accompanying image attention maps for Figure 17.