[Reviews · NeurIPS 2019]

Reviewer 1



- The paper presents a novel approach of getting attention at a given time by modeling infinitesimal change of attention as a continuous function. They extend the idea proposed in "Neural Ordinary Differential Equations" and implement in the MAC framework. They show that this proposed approach can get comparable performance by taking 1/3rd the steps. - I enjoyed reading the paper and detailed experimental analysis. The paper is clearly written and the take-aways from the experiments are clearly outlined. I specially liked the qualitative analysis on interpretability by discussing chunkiness of attention maps, consistency across different seeds and the dynamics of the interpolation between different attention steps. - The paper lacks discussion on the advantages of continuous attention mechanism instead of discrete attention mechanism such as Adaptive Computation Time [1]. Approaches like [1] has also shown reduction in computational complexity / attention steps while preserving performance. - The paper also provides enough technical details (both in supplementary and main manuscript) for easy reproducibility. The code provided should also help in that aspect. - The proposed method should help in building computationally efficient algorithms for visual reasoning. Additionally the proposed metric to measure focus drift for model might also be useful to reason about how the attention changes over time. [1]Alex Graves. Adaptive computation time for recurrent neural networks. arXiv preprint arXiv:1603.08983, 2016. Update [Post Rebuttal] Overall I thought that the paper is of a high quality. I found the idea using Neural ODEs in the MAC framework to model attention as a continuous system is definitely interesting. After going through the discussion and additional experiments in the rebuttal, I believe that the authors gave further evidence on the significance of their work and the generalizability of the approach. Therefore, I am sticking to my original decision.

Reviewer 2



The authors attempt to imitate human cognition, allowing MAC attention to have continuous reasoning. This is achieved by incorporating neural ODEs into the logits calculations per time stamp. This is original idea. I find the writing facaniting, and the overall quality high. I was really excited to hear the paper inspiration is human cognition and continuous reasoning. It is important research goal to take inspiration from human behaviour. The issue with this work is the significance. The model and evaluation relies on a single model(MAC), and not used in other types of attention (e.g., Glimpses, Transformer Heads). This also limits the impact to a specific task (CLEVER), and the overall CLEVER accuracy stays the same. The paper also discuss a new evaluation method(TLT), which finds the amount of attention shifts. But this evaluation can only be tested with MAC-like models, which is not likely to remain a state-of-the-art model.

Reviewer 3



Post-rebuttal Comments: After reading the other reviews, the author's response, and the reviewer discussions, below are my thoughts: (P1) The direction of the paper is definitely interesting. (P2) However, I am not completely convinced that TLT is necessarily a good metric for interpretability. (P3) Given that the benefits of the proposed approach are mostly in TLT and with the above point, it is not clear if this leads to better interpretability. On a side note, I would like to thank the authors for a strong rebuttal -- including experiments on GQA. However, concerns (P2) and (P3) still remain. Therefore, I am sticking to my score. General comments: (G1) The paragraph explaining the relation of TLT to Occam’s Razor (L32-L41) sounds very philosophical without supporting evidence (either through related work or studies). Further, arguments for why TLT is a metric of interpretability is not convincing (L32-L41). There are no further empirical or human studies performed to established this clearly. (G2) How does the system handle discontinuous attentions, where it makes sense. For instance, to answer the question: ‘Are there equal number of cubes and spheres?’, human would (i) find all the cubes (attention_cube), (ii) find all spheres (attention_sphere), and then compare their counts. Intuitively, attention_cube and attention_sphere needs to be discontinuous. From what I understand, the ODE solver discretizes this sudden shift as an interpolation between two time steps (t=1 and t=2). If this is true, isn’t this in disagreement with the proposed idea of continuous attention change across steps? (G3) A major concern is insufficient experimental validation. The proposed approach has similar performance as prior work. Most of the benefits are obtained on the TLT metric. With the problems in (G1), contributions feel not sufficiently backed with empirical evidence. (G4) Additionally, the paper does not contain any experiments on any of the visual question answering (VQA) real datasets. Without these experiments, it unclear if models with proposed attention regularization across steps has benefits on real datasets (and therefore applications). This is another drawback of the current manuscript. (G5) The manuscript does not talk about the additional cost of running the ODE solvers including run-time analysis and comparisons. Typos: L44-45: Grammatical error in the sentence needs a fix.

[Author Response · NeurIPS 2019]

*Thank you all for your thoughtful comments; we address your concerns below.*

**[R3] Clarify relation between TLT, Occam's razor, and interpretability** We discussed this a bit in our paper by drawing connections to Feldman's work (L36), but we agree that the relation between the three topics should be expanded upon. Since TLT measures the total stepwise change in attention weights, decreasing TLT can be seen as an instance of the minimum description length (MDL) principle. The MDL principle formalizes Occam's razor and is a reasonable relaxation of Kolmogorov complexity. Previous work in cognitive neuroscience starting from [Hochberg and McAlister. A quantitative approach, to figural "goodness". *Journal of Experimental Psychology*, 46(5):361, 1953.] to its modern follow-up studies empirically demonstrate that such quantitative complexities are inversely proportional to interpretability. We will add the discussion of such relevant studies to section 1.

**[R2] "In addition, MAC is only tested on the syntactic CLEVER dataset."**
**[R3] "the paper does not contain any experiments on any of the visual question answering (VQA) real datasets"**

| Model(step) | MAC(2) | DAFT MAC(2) | MAC(4) | DAFT MAC(4) | MAC(8) | DAFT MAC(8) |
|---|---|---|---|---|---|---|
| Accuracy | $52.4 \pm 0.3$ | $52.4 \pm 0.7$ | $52.6 \pm 0.1$ | $52.6 \pm 0.4$ | $52.7 \pm 0.5$ | $52.8 \pm 0.5$ |
| TLT | $0.43 \pm 0.09$ | $0.19 \pm 0.14$ | $0.94 \pm 0.23$ | $0.34 \pm 0.08$ | $2.02 \pm 0.23$ | $0.48 \pm 0.05$ |

To show the benefits of DAFT on real-world datasets, we additionally trained and evaluated DAFT MAC on the GQA dataset [Hudson and Manning. Gqa: A new dataset for real-world visual reasoning and compositional question answering. *CVPR*, 2019.]. Due to insufficient time, we trained the models with the 1M subset of the GQA dataset rather than the 14M full training set. We found that DAFT MAC preserves all the benefits: no harm on performance, lower TLT, and consistent and chunked attention maps. Note that that GQA requires less steps than CLEVR, and also that TLT values for GQA are much lower than that for CLEVR (over 4 for MAC(8)). This evidence supports our claim that TLT measures effective reasoning path length. We will add these results and accompanying visualizations to appendix.

**[R3] "The manuscript does not talk about the additional cost of running the ODE solvers including run-time analysis and comparisons."**

| Model (solver) | MAC | DAFT MAC (euler) | DAFT MAC (rk4) | DAFT MAC (dopri5; used in training) |
|---|---|---|---|---|
| Time (ms) | $153.7 \pm 3.8$ (1x) | $167.9 \pm 1.7$ (1.09x) | $189.7 \pm 1.9$ (1.23x) | $365.5 \pm 12.5$ (2.37x) |
| Accuracy | $98.6 \pm 0.2$ | $98.7 \pm 0.2$ | $98.9 \pm 0.2$ | $98.9 \pm 0.2$ |
| TLT | $2.06 \pm 0.15$ | $1.76 \pm 0.07$ | $1.62 \pm 0.06$ | $1.62 \pm 0.06$ |

For more detailed run-time analysis, we measured the time, accuracy, and TLT obtained by using various ODE solvers *during evaluation* of five different 4-step DAFT MAC. We used two fixed-step solvers (Euler method and Runge-Kutta 4th order method with 3/8 rule) and one adaptive-step solver (Dormand-Prince method) that we used during training. We found that during evaluation, rk4 solves all the dynamics generated from CLEVR dataset. Even the simplest euler method results in lower accuracy and higher TLT compared to vanila MAC. We will add these results to section 5.2.

**[R2] What if instead of neural ODE, $a_t$ simply depends on $a_{t-1}$?** As discussed in the previous question on run-time analysis, we have tried the Euler method, which is equivalent to the residual form $a_t = a_{t-1} + f(a_{t-1})$. While using such a simple dependence improved over vanila MAC, this scheme was outperformed by DAFT MAC with more sophisticated solver both in terms of accuracy and TLT.

**[R2] "the approach is novel, but only applicable for MAC-like solution."** We believe that we insufficiently emphasized the generality of DAFT. In fact, DAFT can be applied to any attention update via a one-line change: $\mathbf{a}_{t_1} = \mathbf{W}^{1 \times d}(\mathbf{W}_{t_1}^{d \times d}\mathbf{q} \odot \mathbf{cw}) \rightarrow \mathbf{a}_{t_1} = \mathbf{a}_{t_0} + \int_{t_0}^{t_1} f(\mathbf{a}_t, t)dt$. DAFT can therefore be used in the types of attention that R2 mentions such as glimpses or Transformer heads. To highlight this fact, we will add short pseudocode detailing the application of DAFT to other attention mechanisms to section 4.

**[R2] "...DAFT doesn't improve the performance of MAC. I do tend to believe that low TLT should also correlate with better performance."** The main focus of this paper is not improving performance, but rather on being maximally interpretable while harming performance as little as possible. TLT is highly correlated with the interpretability of the model. This claim is backed by many previous works in cognitive science in addition to our empirical results.

**[R1] "The paper lacks explanation about ODE Solvers used in the method..."** We will add a concise explanation of fixed- and adaptive-step ODE solvers to section 2.3 for self-containedness.

**[R1] "Further discussion and comparison to discrete attention mechanism..."** We will add connections between our method to pondering techniques such as ACT and dynamic halting (of Universal Transformer) to section 2.3.

**[R2] "The qualitative example (Fig. 3) is kind of weird. The attention seems to focus a lot on the edges, and never on green blocks, can you please comment on that?"** In the CLEVR dataset, "blocks" refer to cubes. The image in figure 3 contains no green cubes, so the model is correct in not attending to any other object.

[Meta-Review · NeurIPS 2019]

After considering the author response and discussing the paper, all reviewers agreed that the paper presented interesting and novel approaches to the problem. It is however unclear what advantages the approach provides given the extra complexity and computational burden (over 2.2x due to the ODE solver used). The approach did not significantly affect the accuracy; however, it did result in less jumpy attention that tended to form coherent blocks. This was captured in the proposed TLT metric; however, the submission labels this metric as indicating greater interpretability. It makes general sense that less jumpy attention would lead to greater interpretability but I encourage authors to clarify this point in future revisions (more so even than in the rebuttal). While not revolutionizing attention, this work proposes an interesting direction and delivers a useful measure (TLT) for evaluating stability of attention over time.